# Introduction of Surface Modifiers on the Pt-Based Electrocatalysts to Promote the Oxygen Reduction Reaction Process

**DOI:** 10.3390/nano13091544

**Published:** 2023-05-04

**Authors:** Haibin Wang, Yi Wang, Chunlei Li, Qiuping Zhao, Yuanyuan Cong

**Affiliations:** 1School of Petrochemical Technology, Lanzhou University of Technology, Lanzhou 730050, China; whblut@163.com (H.W.); wangyi@lut.edu.cn (Y.W.); licl@lut.edu.cn (C.L.); zqp_lz@163.com (Q.Z.); 2Key Laboratory of Low Carbon Energy and Chemical Engineering of Gansu Province, Lanzhou University of Technology, Lanzhou 730050, China

**Keywords:** oxygen reduction reaction, Pt-based electrocatalysts, surface modification

## Abstract

The design of Pt-based electrocatalysts with high efficiency towards acid oxygen reduction reactions is the priority to promote the development and application of proton exchange membrane fuel cells. Considering that the Pt atoms on the surfaces of the electrocatalysts face the problems of interference of non-active species (such as OH_ad_, OOH_ad_, CO, etc.), high resistance of mass transfer at the liquid–solid interfaces, and easy corrosion when working in harsh acid. Researchers have modified the surfaces’ local environment of the electrocatalysts by introducing surface modifiers such as silicon or carbon layers, amine molecules, and ionic liquids on the surfaces of electrocatalysts, which show significant performance improvement. In this review, we summarized the research progress of surface modified Pt-based electrocatalysts, focusing on the surface modification strategies and their mechanisms. In addition, the development prospects of surface modification strategies of Pt-based electrocatalysts and the limitations of current research are pointed out.

## 1. Introduction

Proton exchange membrane fuel cells (PEMFCs) are one of the most promising fuel cells due to their high efficiency and also being environment-friendly and energy saving, which are favored by many researchers [1,2,3]. However, the extensive usage of Pt electrocatalysts to accelerate the sluggish oxygen reduction reaction (ORR) kinetics of cathodes has seriously increased the manufacturing cost of PEMFCs [1,4,5,6,7]. It is essential for research to develop high-performance Pt-based electrocatalysts to improve ORR kinetics and reduce the consumption of Pt. The activity of platinum group metals electrocatalysts is usually measured by mass activity (MA, the current density normalized by the mass of the electrocatalysts) because they are priced by mass [8,9]. According to the targets of the US Department of Energy (DOE), the MA of the electrocatalysts for ORR should be more than 440 mA mg_Pt_^−1^ @ 0.9 V_IR-free_ vs. RHE; the loss rate relative to the initial MA should be less than 40% after the long-term durability tests [10,11,12]. To date, great efforts have been devoted to develop Pt-based electrocatalysts by doping non-precious transition metals (Fe, Ni, Cu, etc.) into the Pt lattice and regulating their morphologies, such as core-shell structures, nanowires, heterostructures, polyhedrons, etc. [9,13,14,15,16]. This material-focused research has been widely and reliably presented in both performances and mechanisms [17,18,19,20]. However, inadequate interfacial reaction driven by near-surface environmental degradation (such as the poisoning of surface atoms by some polar anions, competitive adsorption of intermediate species, and sluggish mass transfer) are difficult to overcome with the traditional strategies of optimizing the intrinsic structure of catalysts, which was ignored in many research papers exploring the ORR process. 

With the participation of O_2_, H^+^, and metal atoms in ORR process, the adsorption/desorption of oxygen-containing species (O_ad_, OOH_ad_, OH_ad_) and the formation of H_2_O molecules on the surfaces of Pt-based nanoparticles (NPs) indicate that the ORR process is extremely sensitive to the surface properties of electrocatalysts [21,22,23,24]. Surface engineering is regarded as a reasonable way to enhance the catalytic performance, which introduces foreign species (such as carbon, silicon, organic compounds and ionic liquids) to modify the reaction interfaces of surface Pt atoms [25,26,27,28,29]. Up to now, multiple researchers have proved that the surface modification of Pt electrocatalysts can significantly enhance their ORR performance even though some active sites are inevitably covered [25,30]. Nonetheless, the rational design and revealing action natures of surface modification remain challenges. It is essential to deeply understand the surface modification engineering of Pt based electrocatalysts, which can provide guidance for optimizing the reaction interfaces of electrocatalysts to achieve an optimal ORR performance.

In this review, we summarized the research advances of surface modification engineering toward the acidic ORR process (The types of surface modification and their effects are summarized in Table 1) and presented the structure–activity effects of the electrocatalysts surface modification. Separately, we investigated the mechanisms of surface modification strategies in Pt-based electrocatalysts in terms of corrosion resistance, weakening OH_ad_ adsorption, optimizing mass transfer, and regulating hydrophobicity. The resulting performance characteristics have also been analyzed to gain an in-depth insight into the role of surface modification engineering. Meanwhile, some challenges of Pt-based electrocatalysts in surface modification engineering were proposed in order to prospect the future development.

## 2. Surface Modification with Silicon and Carbon

The surface atoms of Pt-based NPs tend to be dissolved and oxidized in the acid ORR process, which may induce the disintegration and Ostwald ripening of NPs [42,43]. Some researchers have designed cage-like structures with species that are stable with chemical and physical properties to lock Pt-based NPs inside. For example, the Ostwald ripening effect of Pt-based NPs with the coating of a SiO_2_ layer in accelerated durability tests (ADTs) was effectively suppressed [44]. Notably, the pores size on the SiO_2_ has a great influence on mass transfer and conduction for the ORR [45]. Takenakan et al. believe that porous SiO_2_ corresponds to the Si-O-Si bond; they prepared the electrocatalysts of CH_3_-SiO_2_/Pt/carbon black (CB) and SiO_2_/Pt/CB using methyltriethoxysilane (MTEOS) and tetraethoxysilane (TEOS) as porous SiO_2_ precursors, respectively. The electrochemical test results showed that SiO_2_/Pt/CB, using TEOS as the precursor of SiO_2_, had an inhibitory effect on the activity, which was inferior to the performance of CH_3_-SiO_2_/Pt/CB. This observation is attributed to the higher condensation degree of TEOS, resulting in the smaller siloxane ring of SiO_2_ prepared by TEOS, hindering the diffusion of O_2_ and weakening the activity of electrocatalysts. In contrast, the SiO_2_ layer prepared with MTEOS as the precursor is considered to have a siloxane ring of appropriate size, and thus did not inhibit the diffusion of O_2_ and proton conductivity. Moreover, the results of the ADTs indicated that the LSV curves of CH_3_-SiO_2_/Pt/CB had almost no change after 10,000 CV cycles between 0.6 and 1.0 V vs. RHE. Obviously, the SiO_2_ cages prevented the disintegration of Pt-based NPs during ADTs [46]. 

With the consideration of electrical conductivity, Pt-based NPs coated by porous carbon is also a common surface modification strategy, and such carbon layers are generally obtained by pyrolysis and carbonization of organic polymers coated on the electrocatalysts surfaces, as shown in Figure 1a [47]. The N-doped carbon layers on the surfaces of Pt-based NPs are widely researched to improve the stability of electrocatalysts. For one, the formation of N-C causes the π bond in C=C to break, leaving a more stable C-C bond, which enhances the corrosion resistance of the carbon layers. For another, the electronic structure of Pt can be regulated by N atoms and promotes Pt to improve its thermodynamic stability in acidic media. Nie et al. developed the polyaniline (PANI) coating–annealing strategy to prepare N-doped carbon (NC) coated PtNi/C@NC. The X-ray photoelectron spectroscopy (XPS) shows that the binding energy of Pt 4f peaks for PtNi/C@NC shifted positively relative to that of PtNi/C (Figure 1b), indicating that there is electronic interaction between Pt and N atoms, creating stable loading of PtNi NPs [33]. In addition, the relatively high electronegativity of the N-doped carbon layer can contribute to modify the charge transfer and resist the adsorption of poisonous species. For example, in Figure 1c,d, compared with Pt/C and D-Pt-Fe/C, the electrocatalyst of O-Pt-Fe@NC/C with N-doped carbon encapsulating exhibits a lower CO oxidation peak potential and less half-wave potential (*E*_1/2_) decrease in the presence of CO and NaHSO_3_, respectively, indicating a better anti-poisoning property of O-Pt-Fe@NC/C towards CO and SO_3_^2−^ [31]. Zhao et al. insisted that the N-doped carbon layer can improve the polarity and enhance the hydrophilicity of the surface of electrocatalysts, which enables the fuel cell to operate at lower humidity [48]. However, some intermediates such as OH_ad_, OOH_ad_, and O_ad_ tend to interact with water molecules by hydrogen bond, which significantly stabilizes the adsorption of intermediates on Pt atoms [49,50]. These intermediates with the high binding energy on Pt result in the parts of Pt active sites being occupied, which is not conducive to the ORR catalytic process. This problem has attracted more attention in the field of ORR catalysis, and most researchers aim at reducing the adsorption energy of intermediates and regulating the hydrophobicity on electrocatalysts [51,52,53]. 

## 3. Surface Modification of Amine Compounds

Conventionally, it was believed that the attachment of organic molecules to the surfaces of the electrocatalysts would limit the ORR process due to surface coverage and reduced conductivity [38,54]. Therefore, various measures were employed to remove the surfactants and organic reactants after the synthesis of the electrocatalysts to expose enough active sites [55,56]. In recent years, researchers have found that proper adsorption of specific organic ligands or compound molecules on the surfaces of Pt NPs can significantly promote their ORR activity, and many attempts have been applied to modify the surfaces of Pt NPs with organic molecules [57,58].

### 3.1. Research Progress of Amine Compounds for Surface Modification

Given the great affinity of Lewis acid and base interactions on Pt NPs, the amine groups tend to be adsorbed on the surfaces of Pt NPs [58,59]. At present, there have been many studies on amine molecules-modified Pt NPs to promote ORR performance, such as melamine [35,60], aniline, alkyl amine [57], tetra-(tert-butyl)-tetraazaphorylin (tBuTAP) [61], and oleamines (OA) [59]. Their activities are summarized in Table 1. Melamine is the most studied surface modifier because of its rich amine groups and excellent activity promotion (Figure 2a) [58,62,63,64]. For example, Yamazaki groups adsorbed melamine molecules on the Pt/Pd/C surfaces (as shown in Figure 2b) by a facile method (immersing the working electrode into 10 mM melamine solution for ten minutes) [34]. The cyclic voltammetry (CV) curves before and after surface modification are shown in Figure 2c. After melamine decoration, the ECSA of Pt/Pd/C calculated by integrating the area of hydrogen desorption region decreased from 110 to 95 m^2^ g_Pt_^−1^, indicating that 14% of the Pt active sites are occupied by melamine molecules. Moreover, the peak of the formation of Pt–OH_ad_ and the reduction peak of Pt oxide shifted to higher potentials, indicating that the modification of melamine molecules can hinder the oxidation of surface Pt atoms and the adsorption of OH_ad_ species. Additionally, the linear sweep voltammetry (LSV) curves are shown in Figure 2d and demonstrate that the half-wave potential (*E*_1/2_) of the melamine modified Pt/Pd/C increased for 27 mV compared with the unmodified sample. In addition, the calculated MA of the melamine modified Pt/Pd/C is 1.97 times that of bare Pt/Pd/C. 

The adsorption of amine groups of alkyl amine molecules on Pt atoms has also been proved to promote ORR kinetics by optimizing the adsorption energy of reaction intermediates on Pt atoms [30,38,65]. Miyabayashi et al. modified the Pt NPs of Pt/carbon black (CB) with the suitable ratio of pyrene group (PA) and octyl amine (OA) as shown in Figure 3a. The activity of Pt/CB modified by OA/PA (7/3) was significantly improved (as shown in Figure 3b,c). More interestingly, the MA of OA/PA (7/3) Pt/CB only lost 2% after 10,000 CV cycles in the durability test [57]. This is attributed to the amine and aromatic groups in PA molecules interacting with Pt atoms and carbon supports, respectively, to anchor the Pt NPs on the CBs, and thus suppressing migration and aggregation of Pt NPs on carbon black supports [35,66]. Compared with the modification of small molecules on Pt nanoparticles, planar macrocyclic porphyrin compounds can also be used as optional surface modifiers. For example, Co-tBuTAP prepared by coordination of tBuTAP and Co^2+^ can significantly promote the ORR performance after the modification of Pt nanoparticles [36]. As shown in Figure 3d, the XPS spectra peaks of Pt 4f for Pt/C and tBuTAP-adsorbed Pt/C catalyst are no offset from each other [36,61]. This is because only a few planar macrocyclic compounds can be adsorbed on Pt nanoparticles due to their similar size, which results in insufficient induction of the electronic structure of Pt atoms on the surfaces [61]. However, the N atoms on the tBuTAP have a great affinity for the Pt atoms, which makes the Co-tBuTAP molecules stably adsorb on the surfaces of the Pt nanoparticles [36,67].

### 3.2. The Action Mechanism of Amine Compounds 

The mechanism of acidic ORR on Pt is as follows (Equations (1)–(5)), in which O_2_ adsorbed on Pt atoms combines with protons to form intermediates (OOH_ad_, O_ad_, and OH_ad_) [49]. These intermediates with strong chemical polarity are difficult to desorb from Pt NPs surfaces, and thus are considered as performing competitive adsorption of active species, such as O_2_ and H^+^. Especially, desorption of OH_ad_ on Pt is the last step in the ORR process. The OH_ad_ was commonly used to represent the intermediate species for researching. Many researchers revealed that OH_ad_ tends to adsorb on the low coordination Pt atoms located at edges, corners, and steps at high potential (0.6–1.0 V vs. RHE), and thus it was widely believed to inhibit the ORR [60,68].
(1)$$O_{2}+\;*\mathrm{Pt}↔O_{2_{\mathrm{ad}}}$$
(2)$$O_{2_{\mathrm{ad}}}+H^{+}+e^{-}↔{\mathrm{OOH}}_{\mathrm{ad}}$$
(3)$${\mathrm{OOH}}_{\mathrm{ad}}+H^{+}+e^{-}↔O_{\mathrm{ad}}+H_{2}O$$
(4)$$O_{\mathrm{ad}}+H^{+}+e^{-}↔{\mathrm{OH}}_{\mathrm{ad}}$$
(5)$${\mathrm{OH}}_{\mathrm{ad}}+H^{+}+e^{-}↔\;*\mathrm{Pt}+H_{2}O$$

The adsorption of amine compounds on the low coordination Pt atoms can effectively destabilize the OH_ad_ on Pt, especially at high potentials. Daimon et al. modified Pt/Ketjen Black (KB, a kind of carbon black) electrocatalyst with melamine. Obviously, the LSV polarization curve of melamine modified Pt/KB have positively shifted to higher potential compared with the initial Pt/C in the Temkim adsorption region, and the more positive offset was presented with the potential increases (Figure 4a) [35]. Similarly, Yamazaki et al. found that the MA enhancement of melamine modified PtPd/C compared with initial PtPd/C measured at 0.95 V vs. RHE was higher than that at 0.9 V vs. RHE. All of these results suggest that the introduction of the surface modifier of melamine can effectively promote the ORR kinetics, so the LSV curve shows a significant increase in the high potential region. To further investigate the mechanism of ORR promotion, the researchers modified PtPd/C surface with Co-tBuTAP and measured CO stripping voltammograms (Figure 4b). CO is often employed in studying the catalytic mechanism of ORR. The CO molecules adsorbed on the surfaces of Pt NPs were oxidized by the OH_ad_ and then stripped immediately. Therefore, the adsorption state of OH_ad_ on Pt NPs can be analyzed by measuring the strength and position of the stripping peak of CO in the CV curves. The CO stripping peak in the CV curves of Co-tBuTAP modified PtPd/C positively shifted relative to as-synthesized PtPd/C, indicating that the oxidation of CO was limited. Obviously, this indicates that the adsorption of OH_ad_ on Co-tBuTAP modified PtPd/C is inhibited because the CO adsorbed on the electrocatalyst was insufficiently oxidized by lesser amount of OH_ad_ [34]. In addition, the adsorption of amine compounds on Pt based electrocatalysts is attributed to the electronic interaction between amine group and surface Pt atoms, in which N atoms donate electrons to Pt atoms. Therefore, the adsorption of OH_ad_ on surface Pt atoms is weakened with the increased electron density of Pt [38,69,70]. For the mechanism of amine compounds inhibiting the adsorption of OH_ad_ on Pt, the currently accepted explanation is as follows: Orderly and stable water molecular networks are formed on the Pt NPs surfaces without any surfactant during the ORR process, which significantly stabilizes the adsorption of OH_ad_ through a hydrogen bond [49,50]. Conversely, the amine compounds penetrate and destroy the structure of water molecular networks, and thus destabilize the adsorption of OH_ad_ on Pt nanoparticles [60,71]. 

## 4. Ionic Liquid Surface Modification

Water molecules on Pt(111) are connected by hydrogen bonds to form stable and orderly networks, as shown in Figure 5a. The water molecular networks on the surfaces of Pt NPs are the main components of outer Helmholtz layer of the double electric layer (DEL) and the mediums of interfacial reaction [72]. However, these water molecular networks limit ORR kinetics for the following reasons: the stabilizing effect on inactive intermediates by the hydrogen bonding of H_2_O-OH; low O_2_ solubility and mass transfer capacity of water; and insufficient ionic concentration contributing to low proton conductivity of water [50,73,74,75]. According to the research of Mcneary and Nakamura groups, the hydrophobic surface weakens the noncovalent interactions between specifically adsorbed species, not only inhibiting the adsorption stability of the anion layer (such as SO_4_^2−^, PO_4_^3−^) on electrocatalysts but also increasing the mass transfer performance [76,77]. It is essential to introduce optimized surface modifiers to inhibit the accessibility of water molecules on Pt nanoparticles surfaces and replace them with more suitable interfacial reaction mediums. Figure 5b shows the mechanism of hydrophobic species inhibiting the adsorption of oxygen-containing intermediate species on Pt(111) by destroying the water molecular networks on it. 

Ionic liquids (ILs) are liquids composed of ions at room temperature [49,79,80]. The ILs with low melting point and viscosity, high conductivity, ideal oxygen solubility, suitable hydrophobicity, excellent electrochemical and thermal stability are expected to be employed for surface modifiers of Pt-based electrocatalysts [41]. Based on the calculation of DFT and Poisson–Boltzmann, Jinnouchi et al. pointed out that the water molecules are adsorbed on the catalysts’ surfaces as the products of ORR on cathode and these water molecules may also lead to blockage of the active sites [27,81]. Hydrophobic ILs can repel water molecules on the catalysts’ surfaces and replace water molecules as reaction media. As shown in Figure 5c–f, the IL of 1-butyl-3-methylimidazolium bis(trifluoromethanesulfonyl)imide ([C_4_C_1_im][NTf_2_]) decorated Pt/C presents high hydrophobicity with the contact angle of 110°. The [C_4_C_1_im][NTf_2_] also selectively adsorbed on the defect sites, hindering the local adsorption of OH_ad_ and the accessibility of water molecules, which is significant for inhibiting oxidation of the defect sites on Pt crystals [27]. Different from the amine compounds, which adsorb on the Pt NPs’ surfaces to destroy the water molecular networks, ILs promote the ORR kinetics by regulating the hydrophobicity on Pt electrocatalysts [27,82,83]. Hydrophobic groups functioned on electrocatalysts contribute to optimized accessibility of O_2_ and H^+^ on Pt atoms [84]. The local dielectric constant is lowered because of the decreased H_2_O molecules density on electrocatalysts surfaces, which promotes the adsorption of non-polar O_2_ on Pt atoms [85]. Recently, to explore the role of hydrophobic ILs in promoting the ORR, Snyder et al. used electrochemical analysis and spectroscopy to research the inhibition behavior of [MTBD][beti] IL (MTBD: 7-methyl-1,5,7-triazabicyclo [4.4.0]dec-5-ene; beti: bis(perfluoroethylsulfonyl)imide) on interfacial water and the promotion mechanism of [MTBD][beti] on ORR (in Figure 6a). They applied [MTBD][beti] as the surface modifier to decorate the surfaces of single crystals of Pt(111) and Pt(110), respectively, and studied their mechanisms on ORR. As shown in Figure 6b, the MA of Pt(111) and Pt(110) modified with [MTBD][beti] are 2 and 1.5 times as much as that of their initial level. This result was believed to be due to the fact that the ordered water molecular networks are more prone to exist on the Pt(111) than Pt(110), while IL reduced the concentration of water molecules at the metal/electrolyte interfaces, reducing the degree of solvation of the adsorbed species, and thus inhibiting the hydrogen bond networks of OH_ad_ and water molecules. Therefore, IL has a more significant catalytic effect on the ORR process on Pt(111). They further employed the attenuated total reflectance surface-enhanced infrared absorption spectroscopy (ATR-SEIRAS) to explore the state of water at the Au/electrolyte interfaces (Au is believed to be more reliable than Pt in differential spectroscopy.) The specific wave number of hydrogen-bonded icelike water was 3290 cm^−1^, while the non-hydrogen-bonded water had wave numbers of 1600 and 1620 cm^−1^. Obviously, with the presence of IL film on Au surface, all the features related to interfacial water were significantly absent or reduced (Figure 6c,d). This indicates that IL inhibits the adsorption of OH_ad_ on metals stabilized by hydrogen bonds of water molecules [78]. In summary, the main mechanism of interfacial hydrophobic ILs enhancing ORR activity is reducing the coverage of oxygen-containing intermediate species on Pt and increasing the density of active sites.

Additionally, the excellent mass transfer capability of ILs is also beneficial to promoting ORR process, which facilitates the transport of H^+^ and O_2_ near the surfaces of the electrocatalysts and thus releases more active sites. [27,39,86,87,88,89]. Luo et al. deposited Pt NPs on poly (2,5-benzimidazole) (ABPBI) coated carbon nanotubes (CNTs) and modified it with the IL of 1-hexyl-3-methylimidazolium trifluoromethansulfonate to prepare CNT/ABPBI/Pt@IL (Figure 7a). Interestingly, the CV curves of hydrogen adsorption/desorption peaks for CNT/ABPBI/Pt@IL did not decrease significantly compared with that of CNT/ABPBI/Pt, as shown in Figure 7b. This means that the IL of 1-hexyl-3-methylimidazolium trifluoromethansulfonate acts as a mass transfer medium and does not occupy the active sites, which is different from the performance of amine organic molecules. The electrochemical test results show that the MA and SA of CNT/ABPBI/Pt@IL are 1.4 and 1.5 times of CNT/ABPBI/Pt, respectively. The TEM image in Figure 7c shows the cable structure formed by IL wrapped around CNT/ABPBI/Pt, and the authors believed that such IL has high proton conductivity, creating an ideal proton transport path, and thus optimizing the phase interfaces on the surfaces of CNT/ABPBI/Pt@IL [90]. In the ORR process, the Nafion ionomer on the electrocatalyst is responsible for transferring protons to the Pt sites. However, this function cannot be performed on the electrocatalysts with high specific surface area carbon supports because the ionomer cannot penetrate the micropores of carbon layers [91,92]. The ionic conductivity of water molecules on Pt NPs is orders of magnitude smaller than that of ionomer, which hinders the proton transport [93]. Modification of electrocatalysts with hydrophobic ILs with high ionic strength is an effective approach to optimize protons transport [94]. Avid et al. modified the high surface area (HAS) Pt/C with alkyl iminazolium bis(trifluoromethylsulfonyl)imine, which significantly promoted the mass transfer effect of O_2_ and protons, as shown in Figure 7d. On the one hand, the Knudsen diffusion of O_2_ in nanopores increases the collision probability of O_2_ molecules with active sites. On the other hand, when the electrocatalysts is positively charged, it repels the protons in the water due to the thick water|Pt EDLs. However, ILs with high ionic strength can shield the electric field near electrocatalysts, making the EDLs of ILs|Pt thinner than that of water|Pt, which is conducive to the transfer of protons to the Pt sites [41].

## 5. Conclusions

In summary, electrocatalysts surface optimizations with silicon oxide layer, N-doped carbon layers, amine compounds, and ILs are designed to determine the effect of improving acid ORR performance. This appears as a more straightforward strategy compared to the design of intrinsic structures and has met with great success in optimizing interfacial reaction for the ORR process. Specifically, the weakening of the adsorption energy of inactive intermediates (O_ad_, OOH_ad_, OH_ad_) is conducive to improving the utilization of active sites on electrocatalysts. This positive effect may be induced by the electronic structure, hydrophobicity, and mass transfer of electrocatalysts.

However, even though the surface modification engineering of electrocatalysts has been extensively studied, there are still some challenges that cannot be ignored: (1) There is no previous research achieving accurate design of surface modification. One of the problems is that they considered only the enhanced performance and failed to quantitatively research the proportion of action sites and the activity. Additionally, the surface structures of the base electrocatalysts are highly required, and the electrocatalysts with high specific surface area are generally preferred as the surface modification object. (2) Few studies have focused on the simulation calculation toward surface modification promoting ORR, limiting further understanding of deeper mechanisms. (3) Importantly, the action stability of some surface modifiers, such as melamine molecules and organic compounds, needs to be further improved, and the desorption problem caused by organic molecules on the Pt-based NPs’ surfaces in long-term work needs to be further studied.

Nowadays, there are fewer reports on surface modification than other regulation strategies, such as crystal structure optimization, but convenient treatments and significant performance improvement will inevitably make this subject develop rapidly. Therefore, several surface modification strategies were summarized in this review to provide references for acidic ORR catalysis.

## Figures and Tables

**Figure 1 nanomaterials-13-01544-f001:**
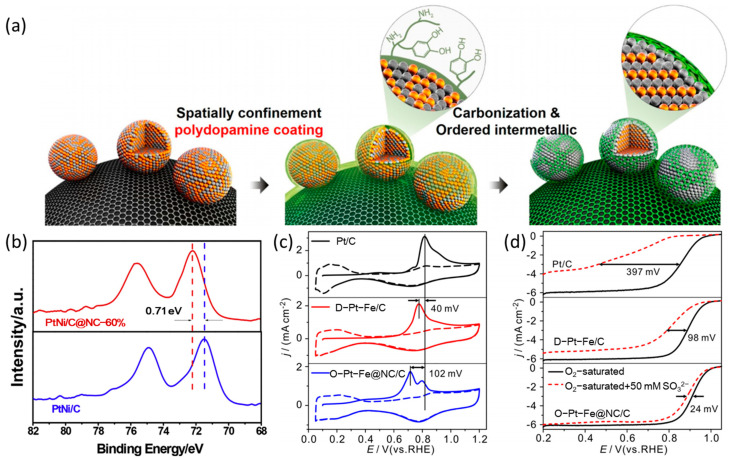
(**a**) Synthesis of carbon coated PtFe/C [47], Copyright © 2015, American Chemical Society; (**b**) the XPS of Pt4f peaks for PtNi/C and PtNi/C@NC-60% catalysts [33], Copyright © 2019, Hydrogen Energy Publications LLC; (**c**) CO stripping curves; and (**d**) polarization curves for ORR in the presence of 50 mM NaHSO_3_ [31], Copyright © 2020, Elsevier B.V.

**Figure 2 nanomaterials-13-01544-f002:**
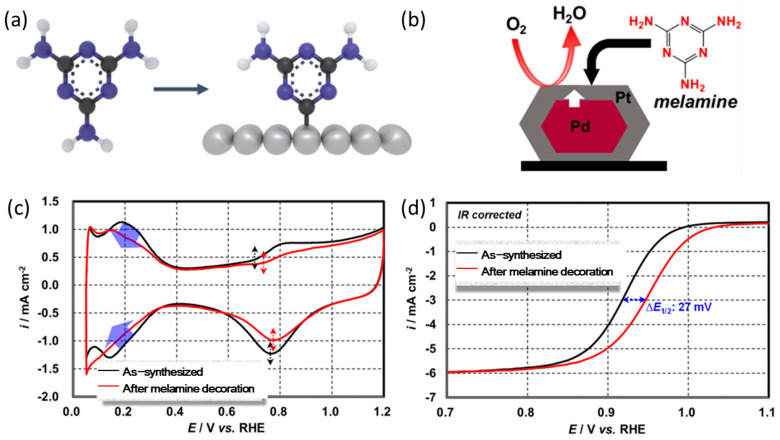
(**a**) The adsorption mode of melamine on Pt(111) surface atoms [59], Copyright © 2021, American Chemical Society; (**b**) the electrocatalyst of Pt/Pd/C modified with melamine, (**c**) CV, and (**d**) LSV of the as-synthesized Pt/Pd/C catalyst (black lines) by melamine decoration with melamine solution (red lines) [34], Copyright © 2020, American Chemical Society.

**Figure 3 nanomaterials-13-01544-f003:**
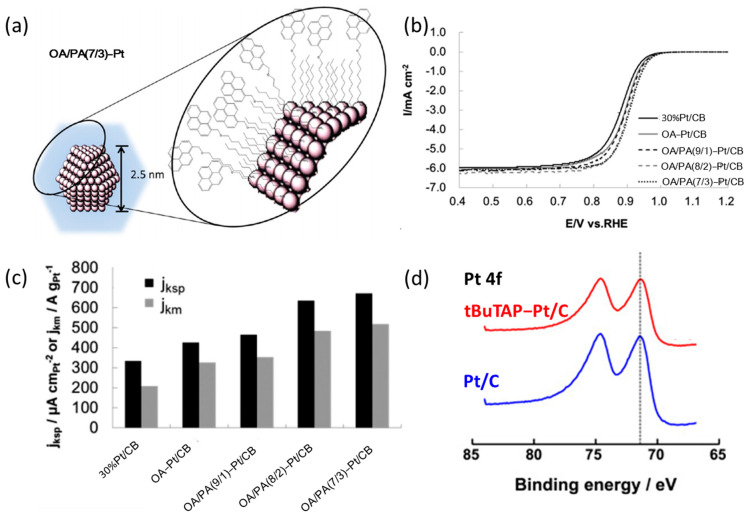
(**a**) Schematic image of OA/PA(7/3)-Pt NP, (**b**) ORR polarization curves for electrocatalysts recorded at room temperature in an O_2_-saturated 0.1 M HClO_4_ aqueous solution with a sweep rate of 10 mV s^−1^ and a rotation rate of 1600 rpm, (**c**) area (j_ksp_) and mass (j_km_) specific activity for the five catalysts at 0.9 V [57], Copyright © 2014, American Chemical Society; (**d**) XPS spectra of Pt/C and tBuTAP-adsorbed Pt/C catalyst [36], Copyright © 2018, Elsevier Ltd.

**Figure 4 nanomaterials-13-01544-f004:**
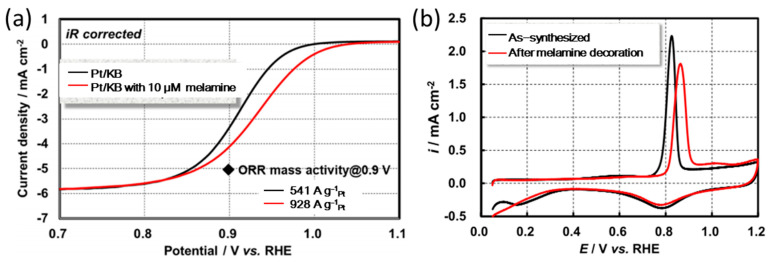
(**a**) LSV curves of Pt/KB measured in O_2_-saturated 0.1 M HClO_4_ at 25 °C, (**b**) CO stripping voltammograms were recorded in 0.1 mol L^−1^ HClO_4_ under Ar-saturated conditions (scan rate: 10 mV s^−1^) at 25 °C after CO adsorption [52], Copyright © 2022, American Chemical Society.

**Figure 5 nanomaterials-13-01544-f005:**
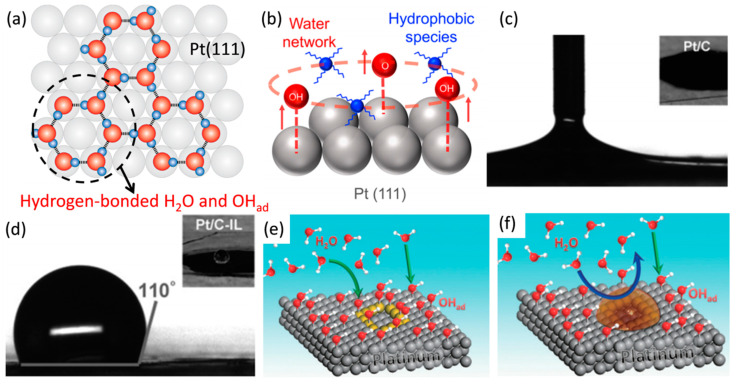
(**a**) Schematic of ordered ice from hydrogen-bonded water structure with OH_ad_ on Pt(111) [78], Copyright © 2023, American Chemical Society; (**b**) the mechanism of hydrophobic species inhibiting the adsorption of O_ad_ and OH_ad_ on Pt(111) [49], Copyright © 2022, American Chemical Society; (**c**,**d**) the hydrophobicity test of Pt/C before and after IL modification, respectively; (**e**,**f**) are the interfacial reaction process of Pt/C before and after IL modification, respectively [27], Copyright © 2016 WILEY-VCH Verlag GmbH & Co. KGaA, Weinheim.

**Figure 6 nanomaterials-13-01544-f006:**
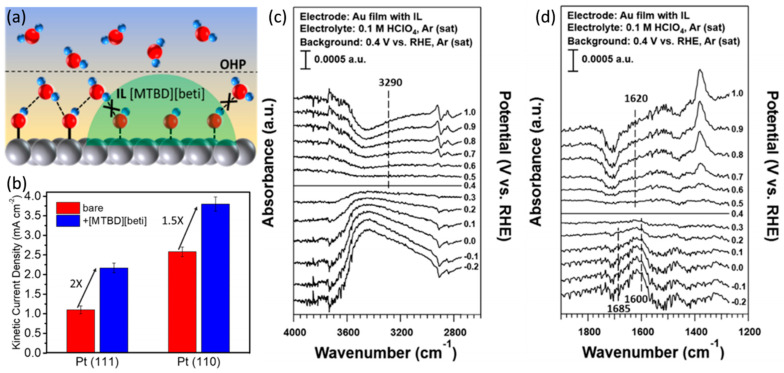
(**a**) Schematic of hydrophobic [MTBD][beti] destabilization of OH_ad_ adsorption on metal surfaces; (**b**) the ORR kinetic current density at 0.9 V vs. RHE for Pt(111) and Pt(110) before and after [MTBD][beti] coating, respectively; (**c**,**d**) the ATR-SEIRAS spectra of the interfacial water on [MTBD][beti] film coated Au electrode in Ar saturated 0.1 M HClO_4_ [78], Copyright © 2023, American Chemical Society.

**Figure 7 nanomaterials-13-01544-f007:**
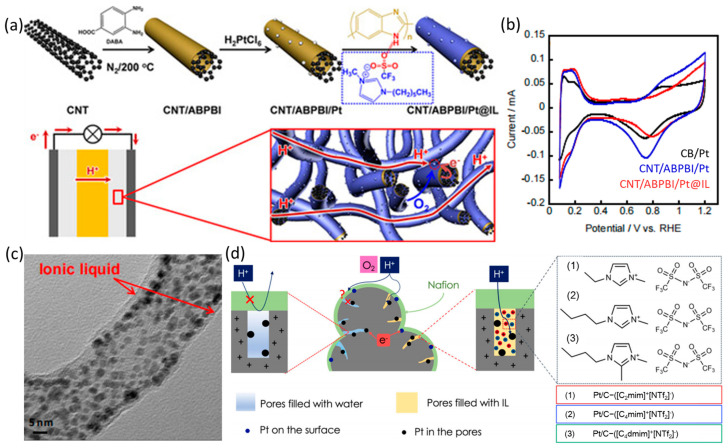
(**a**) Schematic of the CNT/ABPBI/Pt@IL preparation and efficient proton conduction for ORR process; (**b**) CV curves of the electrocatalysts of CB/Pt, CNT/ABPBI/Pt, and CNT/ABPBI/Pt@IL; (**c**) TEM images of CNT/ABPBI/Pt@IL [90], Copyright © 2018 Wiley-VCH Verlag GmbH & Co. KGaA, Weinheim; (**d**) schematic drawing of HSA Pt/C layer interface with the modification of different ILs [41], Copyright © 2022, The Author(s).

**Table 1 nanomaterials-13-01544-t001:** The summary of performances for electrocatalysts with/without surface modifications towards ORR.

Catalysts	Surface Modifiers	Electrolyte	MA (mA mg_Pt_^−1^ @ 0.9 V vs. RHE) before Surface Modification	MA (mA mg_Pt_^−1^ @ 0.9 V vs. RHE) after Surface Modification	Reference
O-Pt-Fe@NC/C	N-doped carbon layer	0.1 M HClO_4_	260	530	[31]
NCM-Pt/C	N-doped carbon matrix	0.1 M HClO_4_	200	360	[32]
PtNi/C@NC	N-doped carbon layer	0.1 M HClO_4_	522	912	[33]
Pt/Pd/C	Melamine	0.1 M HClO_4_	1460 (IR corrected)	3625 (IR corrected)	[34]
Pt/KB	Melamine	0.1 M HClO_4_	541 (IR corrected)	928 (IR corrected)	[35]
PtCo/KB	Melamine	0.1 M HClO_4_	1176 (IR corrected)	1671 (IR corrected)	
Pt/C	tBuTAP	0.1 M HClO_4_	380 (IR corrected)	764 (IR corrected)	[36]
OA/PA-Pt/CB	OA/PA	0.1 M HClO_4_	285	518	[37]
OA-Pt/CB	OA	0.1 M HClO_4_	285	299	
OA/UFHA-Pt/CB	OA/UFHA	0.1 M HClO_4_	204	322	[38]
Pt/C-[C_4_C1im][NTf_2_]	[C_4_C1im][NTf_2_]	0.1 M HClO_4_	330	1010	[27]
fct-PtCo/C@ ILs	[BMIM][TFSI]	0.1 M HClO_4_	470	1040	[39]
PtNiMo/C	[MTBD][BETI]	0.1 M HClO_4_	706	1200	[40]
	[BMIM][NTF_2_]	0.1 M HClO_4_	706	1059	
Pt/C	[C_4_mim][NTf_2_]	0.1 M HClO_4_	288	347	[41]

## Data Availability

No new data were created.

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
