# Peer review of "Introduction of Surface Modifiers on the Pt-Based Electrocatalysts to Promote the Oxygen Reduction Reaction Process"

_nanomaterials, 2023, doi:10.3390/nano13091544_

Round 1

Reviewer 1 Report

English should be revised by a native speaker. At the moment, it is not readable.

1. In  page 2, CB was used without definition. There are some more and all of them should be defined before use.

2. In page 3

A. "and creating stable loading" should be "creating stable loading"

B. "some intermediates.... are tend to..." should be "some intermediates ..... tend to..."

3. In page 3, "Spationally confinemt" shoud be "Spatial confinement"

4. In page 5, "are considered as compeltitive..." this sentence is grammatically incorrect.

Please refer to comments to authors.

Reviewer 2 Report

Report on the manuscript “Introduction of functional species on the surfaces of Pt-based electrocatalysts to promote the oxygen reduction reaction process”, S.-H. Wang et al.

Ref. Nanomaterials_2346503.

General comments:

This is a review manuscript dealing with the functionalization strategies aimed to improve the catalytic efficiency of Pt-based catalysts on the oxygen reduction reaction (ORR). In practice, however, the manuscript is limited to surface modifications with silicon and carbon, amine functionalization and the use of ionic liquids. This is a confusing review needing considerable improvement.

General remarks:

I) The manuscript suffers from the absence of a definite classification of functionalization strategies and even of a clear use of the term functionalization. In principle, this term is of application for systems were the surface of the catalyst is modified via formation of chemical bonds with functional groups. The formation of nanoarchitectures onto the metal surface can be included within this concept, but it is unclear if the coverage of the metal surface by nanodrops of ionic liquids –an adsorption-based process- can be conceived as equivalent. Clarification is needed.

II) The same considerations apply for the case of amines. In pages 4-5, repeatedly the authors mention adsorption of amines on Pt surfaces. This is not exactly a functionalization.

III) The authors emphasize that “ORR is a triple-phase reaction process with the participation of O2, H+ and mental (metal, see below) atoms” (page 2, line 46). This is unclear: in usual voltammetric experiments, O2 (as H+) is in aqueous solution not forming a macroscopic (bubbles) separate phase. Clarification is needed.

III) The authors devote significant space to CO stripping signals (text in page 3 and Figure 1, text in page 6 and figure 4) without the pertinent clarification. Interferes this process on ORR? What is the origin of CO?

IV) The section 3.2 “The action mechanism of amine compounds” reproduces the widely disseminated mechanism of ORR at Pt electrodes in acidic media. As judged by the text in this section, however, the amines exert essentially a role of adsorption mediation. Strictly, no definite mechanistic considerations are described.

Minor remarks:

1) The English should be revised; e.g. page 2, line 46: “metal” rather than “mental” atoms.

Some minor corrections are needed. See comments to authors.

Reviewer 3 Report

The paper is well presented on the important topic of surface functionalization of Pt catalyst in PEM fuel cells to enhance the oxygen reaction reduction. It will make an important contribution to the PEM fuel cell literature.

Round 2

Reviewer 1 Report

Please refer to the comments to authors.

Author Response

Please see the attachment, Thank you!

Reviewer 2 Report

The manuscript can be published in its current version.

Author Response

Your comments will help improve the readability of our manuscript. Thank you again for your review!